# AudioMoG: Guiding Audio Generation with Mixture-of-Guidance

## Abstract

Guidance methods have demonstrated significant improvements in cross-modal audio generation, including text-to-audio (T2A) and video-to-audio (V2A) generation. The popularly adopted method, classifier-free guidance (CFG), steers generation by emphasizing condition alignment, enhancing fidelity but often at the cost of diversity. Recently, autoguidance (AG) has been explored for audio generation, encouraging the sampling to faithfully reconstruct the target distribution and showing increased diversity. Despite these advances, they usually rely on a single guiding principle, *e.g.*, condition alignment in CFG or score accuracy in AG, leaving the full potential of guidance for audio generation untapped. In this work, we explore enriching the composition of the guidance method and present a mixture-of-guidance framework, AudioMoG. Within the design space, AudioMoG can exploit the complementary advantages of distinctive guiding principles by fulfilling their *cumulative benefits*. With a reduced form, AudioMoG can consider parallel complements or recover a single guiding principle, without sacrificing generality. We experimentally show that, given the same inference speed, AudioMoG approach consistently outperforms single guidance in T2A generation across sampling steps, concurrently showing advantages in V2A, text-to-music, and image generation. These results highlight a "free lunch" in current cross-modal audio generation systems: higher quality can be achieved through mixed guiding principles at the sampling stage without sacrificing inference efficiency. Demo samples are available at: `audiomog.github.io`.

## 1 Introduction

Audio generation conditioned on text and video information, known as text-to-audio (T2A) and video-to-audio (V2A) generation, has witnessed significant advancements in recent studies. Typically, these systems generate an audio latent in a small space compressed from the audio waveform or the mel-spectrogram, indicated by the learned text embeddings (Kreuk et al., 2022; Liu et al., 2023; 2024a; Huang et al., 2023b; Evans et al., 2024) or encoded video representations (Xu et al., 2024; Wang et al., 2024a; Du et al., 2023; Luo et al., 2023). Recent efforts have enhanced cross-modal audio generation quality through various perspectives, such as data augmentation (Huang et al., 2023b;a), condition information (Jeong et al., 2024; Wang et al., 2024b; Liu et al., 2023; Li et al., 2024a; Evans et al., 2024), generative models (Kreuk et al., 2022; Liu et al., 2023; 2024a), network architecture (Huang et al., 2023a; Evans et al., 2024; Hung et al., 2024), and compression networks (Liu et al., 2023; Evans et al., 2024; 2025). However, most of these improvements require retraining the model from scratch or with significant overhead.

At the sampling stage, guidance methods have proven effective in enhancing the overall audio generation quality, where classifier-free guidance (CFG) (Ho & Salimans, 2022) is popularly adopted in modern cross-modal audio generation systems (Liu et al., 2023; 2024a; Cheng et al., 2025). By emphasizing the indication of condition signals, namely text or video, CFG can improve the audio generation results under an appropriate guidance scale, while it may therefore sacrifice generation diversity. Recently, autoguidance (AG) (Karras et al., 2024a) is proposed in class-conditioned image generation and has been extended to audio generation by ETTA (Lee et al., 2025). Different from CFG strengthening condition alignment, it guides the diffusion model with a weaker version, encouraging the generation process to faithfully reconstruct the target distribution.

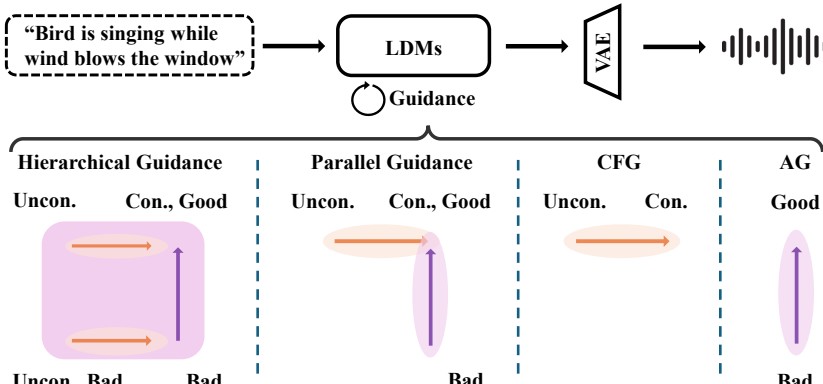

Figure 1: **Overall framework of our proposed AudioMoG,** which illustrates the mechanism of AudioMoG and its degraded forms—Hierarchical Guidance exploits cumulative advantages from both methods for optimal performance, Parallel Guidance introduces complementary directions, and CFG or AG provides a single-directional guidance.

While these guidance methods are advantageous over unguided diffusion sampling, they usually strengthen the generation direction with a single guiding principle, *e.g.*, condition alignment in CFG and score accuracy in AG. It remains underexplored if stronger generation results can be achieved by mixing distinctive guidance methods, while maintaining inference efficiency. In this work, we explore enriching the composition of the guidance method and present a mixture-of-guidance framework, named AudioMoG, to simultaneously consider distinctive guiding principles rather than depend solely on one of them. Firstly, we revisit the design of guidance strategies in audio generation, where we analyze the behaviors and limitations of the widely-used CFG and recent AG, respectively. We demonstrate that, CFG enhances synthesis quality through an entangled effect of score correction and condition alignment amplification, which complicates independent control over quality and diversity—particularly as improvements in the unconditional model diminish the correction signal. In comparison, AG employs a weaker conditional model to isolate the score correction effect, achieving more accurate score estimation to enable quality improvements, though its effectiveness can be sensitive to the choice of the weak model (Karras et al., 2024a; Lee et al., 2025).

Based on these insights, we demonstrate the mechanism of AudioMoG, an improved sampling framework that can fully exploit the complementary advantages of diverse guidance methods. Within the design space, AudioMoG can fulfill the cumulative advantage by progressively harnessing the strengths of diverse guiding principles, reaching the performance of further refining condition-aligned term empowered by CFG with AG, or strengthening both conditional and unconditional score estimation results with AG before CFG. As a degraded form, AudioMoG can fulfill parallel complements or ultimately recover the guidance method considering a single principle, such as CFG or AG, without sacrificing the generality as a mixture framework. Especially, in AudioMoG, we empirically observe that the bad version of the model can be trained using the same network architecture as the good version but with fewer iterations, or even taken directly from earlier checkpoints, avoiding the dedicated design proposed in AG (Karras et al., 2024a) or the sensitivity for audio generation mentioned in ETTA (Lee et al., 2025). Our contributions are summarized as follows.

- We present a mixture-of-guidance framework for audio generation, achieving improved synthesis quality while maintaining the inference efficiency in comparison with the single guiding principle.

- Within the design space, AudioMoG can fulfill the cumulative advantages of diverse guidance methods, progressively expressing their strengths, as well as allowing parallel complements or a single principle as a degraded form.

- Experimental validation on diverse cross-modal audio generation and image generation tasks demonstrates that under the same inference speed, AudioMoG consistently outperforms both CFG and AG. Compared to CFG, we improve FAD from 1.76 to 1.38 in T2A (Evans et al., 2024), from 0.73 to 0.68 in V2A, and from 2.36 to 1.92 in text-to-music generation. Compared to AG, we improve FID from 1.60 to 1.47 in image generation (Karras et al., 2024b).

## 2 PRELIMINARIES

In this section, we introduce the foundation of latent diffusion models and guided audio generation.

**Diffusion-based audio generation system.** In T2A and V2A generation systems, audio signals $x$ are first compressed into a small latent space $z$ with a compression network. Then, latent diffusion models are popularly adopted to learn the generation of audio latent from a simple prior distribution, *e.g.*, the standard Gaussian noise distribution $\mathcal{N}(\mathbf{0}, \boldsymbol{I})$, conditioned on the text prompt or video input. At the training stage, a forward process is introduced to transform the audio latent at $t = 0$ into a noisy latent with

$$z_t = \sqrt{\bar{\alpha}_t} z_0 + \sqrt{1 - \bar{\alpha}_t} \boldsymbol{\epsilon}, \tag{1}$$

where $\sqrt{\bar{\alpha}_t}$ is predefined to control the signal-to-noise ratio in forward process; $\boldsymbol{\epsilon} \sim \mathcal{N}(\mathbf{0}, \boldsymbol{I})$ is the added Gaussian noise and shares the same distribution with the prior distribution $p(z_T)$ at $t = T$. At each training iteration, a noise predictor is optimized with

$$\arg \min_\theta \mathbb{E}_{(z_0, c), \boldsymbol{\epsilon}} \| \boldsymbol{\epsilon} - \boldsymbol{\epsilon_\theta} (z_t, t, c) \|_2^2, \tag{2}$$

where $c$ is the text embedding or encoded video features to indicate audio generation. Given the well-trained noise predictor, a reverse process iteratively reconstructs the audio latent from the prior distribution with

$$p_\theta(z_{0:T-1} | z_T, c) = p(z_T) \prod_{t=1}^{T} p_\theta(z_{t-1} | z_t, c). \tag{3}$$

In sampling, each reverse transition $p_\theta(z_s | z_t, c)$ at the time steps $0 \leq s < t \leq T$ follows a Gaussian distribution $\mathcal{N}(z_s, \mu_{s|t}(z_t, t, c), \sigma_{s|t}^2 \boldsymbol{I})$. The mean and variance are parameterized as

$$\mu_{s|t}(z_t, t, c) = \sqrt{\bar{\alpha}_{s|t}}(z_t - \frac{1 - \bar{\alpha}_{t|s}}{\sqrt{1 - \bar{\alpha}_t}} \boldsymbol{\epsilon_\theta}(z_t, t, c)), \quad \sigma_{s|t}^2 = (1 - \bar{\alpha}_{t|s}) \frac{1 - \bar{\alpha}_s}{1 - \bar{\alpha}_t}, \tag{4}$$

where $\bar{\alpha}_{t|s} = \bar{\alpha}_t / \bar{\alpha}_s$. Given sufficient sampling steps, audio latent $z_0$ is reconstructed and then decoded into audio signals $x$ with a decoding system.

**Classifier-Free Guidance.** CFG (Ho & Salimans, 2022) is one of the most commonly used strategies in diffusion models for conditional generation. During training, the conditional signal $c$ is randomly replaced with the null condition $\varnothing$ with a fixed probability $p_{\text{uncond}}$ (*a.k.a.*, random label dropout), allowing the model to learn both conditional and unconditional noise predictors, $\boldsymbol{\epsilon_\theta}(z_t, t, c)$ and $\boldsymbol{\epsilon_\theta}(z_t, t)$. At inference time, CFG combines the two predictors as follows:

$$\boldsymbol{\epsilon}_{\text{CFG}}(z_t, t, c) = \boldsymbol{\epsilon_\theta}(z_t, t) + w \left( \boldsymbol{\epsilon_\theta}(z_t, t, c) - \boldsymbol{\epsilon_\theta}(z_t, t) \right), \tag{5}$$

where $w \geq 1$ denotes the guidance scale that adjusts the strength of the conditional signal. When $w = 1$, CFG recovers the conditional diffusion model $\boldsymbol{\epsilon_\theta}(z_t, t, c)$. Larger values of $w$ encourage samples to align more closely with the conditioning signal, potentially enhancing generation quality.

**Autoguidance.** AG (Karras et al., 2024a) proposes guiding a diffusion model using a weaker version of itself:

$$\boldsymbol{\epsilon}_{\text{AG}}(z_t, t, c) = \boldsymbol{\epsilon}_{\theta_{\text{bad}}}(z_t, t, c) + w \left( \boldsymbol{\epsilon_\theta}(z_t, t, c) - \boldsymbol{\epsilon}_{\theta_{\text{bad}}}(z_t, t, c) \right), \tag{6}$$

where $\theta_{\text{bad}}$ refers to the weak model with smaller size or less training, and $c$ can be replaced with $\varnothing$. The underlying motivation stems from the observation that the score-matching objective in diffusion models promotes mode coverage, often leading to noisy or inaccurate estimates. By contrastive amplification of the difference between a strong and weak model, AG seeks to improve the score estimation quality. Following a similar rationale, recent works (Kasymov et al., 2024; Phunyaphibarn et al., 2025; Zhong et al., 2025) guide the fine-tuned model using the pre-fine-tuned one. While the specific formulations differ, the core principle remains consistent: leveraging the weak-strong discrepancy to guide improvement.

## 3 AUDIOMOG

### 3.1 ANALYSIS

**2D toy example.** We first provide an analysis of the CFG and AG methods to illustrate their respective guiding principle. We adopt a 2D toy example introduced in Karras et al. (2024a), where a small

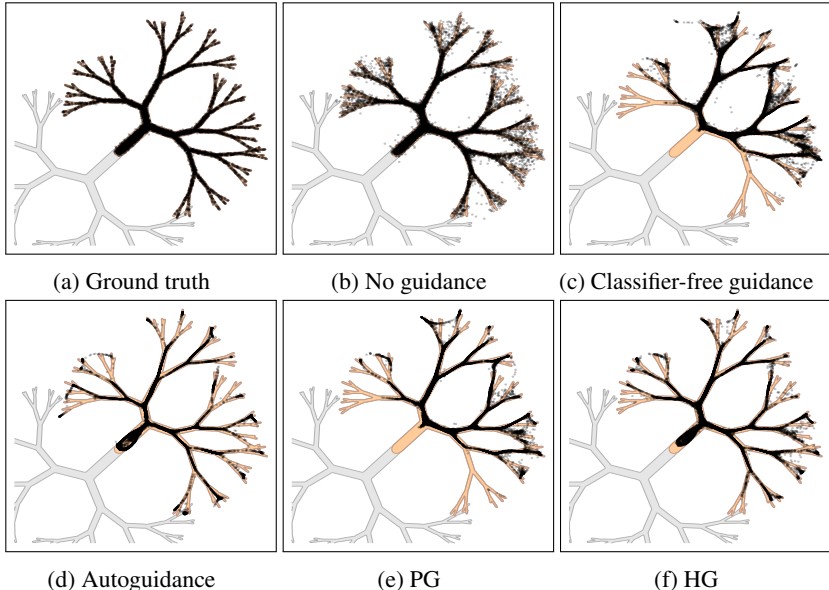

(a) Ground truth      (b) No guidance      (c) Classifier-free guidance

(d) Autoguidance      (e) PG      (f) HG

Figure 2: **Illustration of guidance methods on the fractal-like 2D distribution from Karras et al. (2024a). (a)** Ground truth distribution (orange class). **(b)** Unguided conditional sampling generates outliers. **(c)** Classifier-Free Guidance ($w = 3$) with a well-trained unconditional model struggles to remove outliers. **(d)** Autoguidance ($w = 3$) improves score estimation and removes outliers without reducing diversity. **(e)** Parallel Guidance exhibits mode dropping similar to CFG. **(f)** Hierarchical Guidance eliminates outliers and provides more controllable condition alignment.

denoiser is trained on synthetic data to learn conditional diffusion. The 2D dataset is designed to exhibit low local dimensionality, characterized by highly anisotropic and narrow support, as well as a hierarchical emergence of local detail as shown in Figure 2a, mimicking real-world data manifolds (Karras et al., 2024a). As shown in Figure 2b, the denoiser network learns a suboptimal score function, leading to scattered and unlikely outliers under unguided generation, *i.e.*, conditional diffusion sampling. Additional details on the experimental setup are provided in Appendix E.

**CFG effects.** For guided generation results, CFG improves sample quality by contrasting the conditional and unconditional models. The unconditional model is arguably relatively under-trained due to the inherent difficulty of the unconditional task and the low label dropout rate (*e.g.*, 10% for audio generation in Liu et al. (2023); Deepanway et al. (2023); Evans et al. (2024)). Consequently, the CFG effect in Equation 5 is mixed, as formulated in the following guidance decomposition:

$$\nabla_{\boldsymbol{x}} \log p(\boldsymbol{x}|\boldsymbol{c}) - \nabla_{\boldsymbol{x}} \log p(\boldsymbol{x}|\varnothing) = [\nabla_{\boldsymbol{x}} \log \mathbb{E}_c p(\boldsymbol{x}|\boldsymbol{c}) - \nabla_{\boldsymbol{x}} \log p(\boldsymbol{x}|\varnothing)] + \nabla_{\boldsymbol{x}} \log p(\boldsymbol{c}|\boldsymbol{x}). \quad (7)$$

Namely, with an under-trained unconditional model, the CFG direction entangles two components: (1) *score correction* from weak-strong contrast that eliminates dispersed outliers, and (2) *condition alignment amplification* that may skew the distribution and reduce diversity. The quality improvement observed with CFG can be ideally attributed to the first factor. However, the entanglement makes it difficult for CFG to independently control diversity and quality, as a better unconditional model yields weaker signals for score correction. This effect is visualized in Figure 2c, where CFG with a sufficiently trained unconditional model fails to eliminate dispersion.

**AG effects.** On the other hand, AG avoids this entanglement by employing a weaker but still conditional version of the model, thus isolating the improvement direction without losing diversity as shown in Figure 2d. However, its effectiveness hinges on the availability of a suitably degraded weak model. Constructing such a model in practice can be nontrivial (Karras et al., 2024a; Lee et al., 2025; Jeon, 2025; Hyung et al., 2025), especially when model degradation does not align well with real score estimation errors and the weak-strong contrast cannot provide meaningful directions. In such cases, incorporating additional sources of guidance may be necessary to achieve more robust quality gains. For example, when training data quality is suboptimal, CFG often yields sharper and more prompt-consistent generations due to its "lower-temperature" behavior (Bradley & Nakkiran, 2024), creating a more favorable distribution that can complement AG.

**Motivation.** As discussed above, previous guidance methods, CFG and AG, guide the sampling process with diverse principles, both showing advantages over unguided generation. CFG enhances consistency with conditional information, and AG removes dispersion by mitigating errors. The effectiveness of CFG has been extensively validated in audio generation across various data representations (Liu et al., 2023; Evans et al., 2025), conditional modalities such as text (Deepanway et al., 2023; Huang et al., 2023a), video (Luo et al., 2023; Xu et al., 2024), and network architectures (Liu et al., 2023; Evans et al., 2025; Li et al., 2024a; Hung et al., 2024). However, CFG can still yield suboptimal results due to its overemphasis on condition information. Particularly, it may miss relevant sound events or fail to accurately generate uncommon audio events. Recent work ETTA (Lee et al., 2025) has explored AG on T2A generation, showing increased generation diversity but observing strong sensitivity to the choice of weak model. Given that either CFG or AG has shown quality improvement for audio generation, while both consider a single guiding principle, we explore a mixture-of-guidance framework, aiming at composing guidance methods to fulfill stronger results by exploiting their complementary advantages, *e.g.*, cumulative benefits, even without sacrificing sampling efficiency.

### 3.2 FRAMEWORK

**General setting.** AudioMoG presents a mixture-of-guidance strategy as follows, involving $M$ guidance methods:

$$\boldsymbol{\epsilon}_{\text{MoG}}(\boldsymbol{z}_t, t, \boldsymbol{c}) = \sum_{i=1}^{N} w_i \boldsymbol{\epsilon}_i(\boldsymbol{z}_t, t, \boldsymbol{c}), \quad \text{s.t.} \sum_{i=1}^{N} w_i = 1, \tag{8}$$

where $\boldsymbol{\epsilon}_i$ is a denoiser network and $w_i \in \mathbb{R}$ is the corresponding weight. When $M = 1$, AudioMoG considers a single guidance method, which extrapolates two denoising results as mentioned in (Karras et al., 2024a). Given $M \geq 2$, AudioMoG starts exploiting the complementary advantages of different guidance methods. However, this inevitably increases the complexity, which may result in longer inference time, as the framework considers both the terms required by different methods and the additionally produced interaction terms, *e.g.*, the *unconditional and bad term* when combining CFG and AG. Hence, even though a MoG framework may yield stronger performance by considering more guiding principles and leveraging their complementary benefits, an essential consideration is the balance between synthesis quality and inference speed.

To improve guided audio generation, we empirically observe that combining two distinctive guidance methods, both of which have proven more advantageous than unguided conditional sampling, shows potential to achieve improved synthesis quality without sacrificing inference speed.

**Hierarchical guidance.** When considering a mixture of two guidance methods, the MoG framework shown by Equation 8 exploits their complementary advantages by linearly combining four noise predictors (*i.e.*, $M = 2$, $N = 4$), which can be viewed as hierarchically combining two guidance methods (denoted by HG). Taking CFG and AG as examples for guided audio generation, AudioMoG can be interpreted from two perspectives [1]. Firstly, we can interpret it as refining the CFG method with the AG guiding principle as follows:

$$\begin{aligned}
\boldsymbol{\epsilon}_{\text{CFG}}(\boldsymbol{z}_t, t, \boldsymbol{c}) &= \boldsymbol{\epsilon}_{\boldsymbol{\theta}}(\boldsymbol{z}_t, t) + w_1(\boldsymbol{\epsilon}_{\boldsymbol{\theta}}(\boldsymbol{z}_t, t, \boldsymbol{c}) - \boldsymbol{\epsilon}_{\boldsymbol{\theta}}(\boldsymbol{z}_t, t)), \\
\boldsymbol{\epsilon}_{\text{badCFG}}(\boldsymbol{z}_t, t, \boldsymbol{c}) &= \boldsymbol{\epsilon}_{\boldsymbol{\theta}_{\text{bad}}}(\boldsymbol{z}_t, t) + w_2(\boldsymbol{\epsilon}_{\boldsymbol{\theta}_{\text{bad}}}(\boldsymbol{z}_t, t, \boldsymbol{c}) - \boldsymbol{\epsilon}_{\boldsymbol{\theta}_{\text{bad}}}(\boldsymbol{z}_t, t)), \\
\boldsymbol{\epsilon}_{\text{HG}}(\boldsymbol{z}_t, t, \boldsymbol{c}) &= \boldsymbol{\epsilon}_{\text{badCFG}}(\boldsymbol{z}_t, t, \boldsymbol{c}) + w_3(\boldsymbol{\epsilon}_{\text{CFG}}(\boldsymbol{z}_t, t, \boldsymbol{c}) - \boldsymbol{\epsilon}_{\text{badCFG}}(\boldsymbol{z}_t, t, \boldsymbol{c})).
\end{aligned} \tag{9}$$

Namely, the good and bad terms under the AG framework (Karras et al., 2024a) have been first enhanced by explicitly emphasizing condition alignment, and then AG further strengthens the result towards faithfully reconstructing the target distribution. Alternatively, as proven in Appendix A, it can be interpreted as applying the CFG method to more accurate score estimation results. Namely, the conditional and unconditional terms under the CFG framework (Ho & Salimans, 2022) have been first improved by AG to achieve higher score accuracy. This naturally outperforms previous works that solely depend on CFG: applying CFG to scores not yet refined by AG.

**Other forms: parallel guidance, CFG, and AG.** By controlling the weighting strategy for Equation 8, AudioMoG can show other forms exhibiting a different mechanism when exploiting the

---

[1]HG in CFG-AG or AG-CFG orders yields equivalent guidance family, as proven in Appendix A.

Table 1: **Objective metrics for text-to-audio generation on AudioCaps test set.** The best performance for each metric is highlighted in bold, while the second-best is marked with an underline.

| Model | FAD $\downarrow$ | KL $\downarrow$ | IS $\uparrow$ | FD $\downarrow$ | CLAP $\uparrow$ |
|---|---|---|---|---|---|
| GT | / | / | / | / | 0.52 |
| AudioGen (Kreuk et al., 2022) | 3.13 | 2.09 | / | / | / |
| AudioGen-Large (Kreuk et al., 2022) | 1.82 | 1.69 | / | / | / |
| Make-An-Audio (Huang et al., 2023b) | 1.61 | 1.61 | 7.29 | 18.32 | / |
| TANGO-AF&AC-FT-AC (Kong et al., 2024) | 2.54 | / | 11.04 | **17.19** | / |
| AudioLDM-Large-Full (Liu et al., 2023) | 1.96 | 1.59 | 8.13 | 23.31 | 0.43 |
| AudioLDM 2 (Liu et al., 2024a) | 2.09 | 1.79 | 8.14 | 26.44 | 0.50 |
| AudioLDM 2-Large (Liu et al., 2024a) | 1.89 | 1.54 | 8.55 | 26.18 | 0.53 |
| Stable Audio Open (Evans et al., 2025) | / | 2.14 | / | / | 0.35 |
| CFG-only, $w = 7$ | 1.76 | **1.44** | 13.46 | 20.94 | **0.54** |
| MoG-PG, $w_1 = 4.6, w_2 = 0.2$ | 1.54 | 1.47 | 13.47 | 18.50 | 0.53 |
| MoG-HG, $w_1 = 4.0, w_2 = 3.3, w_3 = 1.2$ | **1.38** | **1.44** | 13.58 | 18.87 | **0.54** |

Table 2: **Subjective metrics for text-to-audio generation on AudioCaps samples.** Rated for overall quality and text relevance, with higher scores indicating better performance.

| Metric | GT | AudioLDM | AudioLDM 2 | CFG-only | MoG-HG |
|---|---|---|---|---|---|
| OVL $\uparrow$ | $3.23 \pm 0.58$ | $2.26 \pm 0.53$ | $2.76 \pm 0.47$ | $3.20 \pm 0.51$ | $\mathbf{3.64 \pm 0.48}$ |
| REL $\uparrow$ | $3.43 \pm 0.62$ | $2.34 \pm 0.61$ | $2.90 \pm 0.55$ | $3.40 \pm 0.59$ | $\mathbf{3.90 \pm 0.54}$ |

complementary advantages. When the interaction terms in Equation 8 are removed (*i.e.*, $M = 2$, $N = 3$), AudioMoG degrades to employing CFG and AG in *parallel* (denoted by PG), which can be written as:

$$\epsilon_{\text{PG}}(z_t, t, c) = \epsilon_{\theta}(z_t, t) + w_1(\epsilon_{\theta}(z_t, t, c) - \epsilon_{\theta}(z_t, t)) + w_2(\epsilon_{\theta}(z_t, t, c) - \epsilon_{\theta_{\text{bad}}}(z_t, t, c)). \quad (10)$$

While PG qualitatively demonstrated by (Karras et al., 2024a) offers a simple and effective integration of CFG and AG, it implicitly assumes their compatibility at each sampling step. However, as discussed above, these two methods guide the model in potentially interfering directions: emphasizing the condition can lead to mode collapse, while correcting from weak references may introduce semantic drift. Thus, it may not be able to fully exploit their complementary advantages in comparison with exploiting the cumulative benefits, as demonstrated by experiments in Section 4.

It is also worth noting that, when further removing the terms in Equation 8, AudioMoG can recover the guidance method considering a single guiding principle, *e.g.,* CFG or AG, indicating that MoG can be seen as a *unified* framework to incorporate diverse guidance methods.

## 4 EXPERIMENTS

### 4.1 T2A EXPERIMENT SETUP

**Datasets.** We use AudioSet (Gemmeke et al., 2017), FSD50k (Fonseca et al., 2021) and Clotho v2 (Drossos et al., 2020). To maintain consistency throughout the dataset, each track in these databases was segmented into 10-second clips and resampled at 16 kHz. The details of these datasets are further introduced in the Appendix F.1. To compare with prior work, we evaluated our models on the widely used AudioCaps benchmark (Kim et al., 2019), which consists of about 1K 10-second audio clips.

**Model configurations.** We use FLAN-T5(Chung et al., 2024) as the text encoder for our base model, and we train a Variational Autoencoder (VAE) (Kingma et al., 2013) that compresses the original waveform into the latent representation. A more detailed description of the model configurations and compression networks is provided in the Appendix F.3. We trained the model for 1 M iterations with a batch size of 8 per GPU. We used the AdamW optimizer (Loshchilov & Hutter, 2017) with a learning rate of 5e-5 and a condition drop $p_{\text{uncond}} = 0.1$ for CFG. The bad model was trained under the same configuration as the main model, but with only 0.1 M iterations. During inference, we use DPM++ 2M SDE (Lu et al., 2022).

**Evaluation metrics.** We conduct a comprehensive evaluation of our models using both objective and subjective evaluations to assess audio generation quality, text-audio alignment, and inference

Table 3: **Objective metrics for video-to-audio generation on VGGSound test set.**

| Model | FAD ↓ | KL ↓ | IS ↑ | FD ↓ | IBS ↑ | Align acc ↑ |
|---|---|---|---|---|---|---|
| GT | / | / | / | / | 32.9 | 83.6 |
| IM2WAV (Sheffer & Adi, 2023) | 6.41 | 2.54 | / | / | 19.0 | 74.3 |
| Diff-Foley (Luo et al., 2023) | 5.79 | 3.12 | 10.8 | 21.90 | 20.4 | **89.9** |
| FoleyGen (Mei et al., 2024) | 1.65 | 2.35 | / | / | 26.1 | 73.8 |
| VTA-LDM (Xu et al., 2024) | 2.01 | 2.37 | 10.4 | 12.80 | 26.2 | 77.0 |
| FoleyCrafter (Zhang et al., 2024) | 2.32 | 2.54 | 9.9 | 18.10 | 27.7 | 83.6 |
| V2A-Mapper (Wang et al., 2024a) | 0.90 | 2.68 | 12.5 | 8.35 | 22.4 | 78.3 |
| VAB-Encodec (Su et al., 2024) | 2.69 | 2.58 | / | / | / | / |
| VATT w/o text (Liu et al., 2024b) | 2.35 | 2.25 | / | / | / | 82.8 |
| CFG-only, $w = 3$ | 0.73 | 2.28 | 17.1 | 4.48 | 32.8 | 85.8 |
| MoG-PG, $w_1 = 2.7, w_2 = 0.15$ | 0.70 | 2.22 | 16.8 | 4.14 | 32.9 | 86.1 |
| MoG-HG, $w_1 = 2.7, w_2 = 2.5, w_3 = 1.2$ | **0.68** | **2.20** | **17.2** | **4.06** | **33.1** | 86.6 |

Table 4: **Comparison between different guidance methods on T2A.**

| Method | FAD ↓ | KL ↓ | IS ↑ | FD ↓ |
|---|---|---|---|---|
| No guidance | 7.31 | 2.45 | 5.86 | 38.42 |
| CFG-only | 1.96 | 1.91 | 7.47 | **17.40** |
| AG-only | 2.30 | 1.91 | 7.41 | 17.57 |
| MoG-PG | 1.67 | 1.54 | 13.52 | 19.01 |
| MoG-HG | **1.38** | **1.44** | **13.58** | 18.87 |

Table 5: **Comparison between different guidance methods on V2A.**

| Method | FAD ↓ | KL ↓ | IS ↑ | FD ↓ |
|---|---|---|---|---|
| No guidance | 1.28 | 2.49 | 10.2 | 8.06 |
| CFG-only | 0.74 | 2.31 | 15.8 | 5.17 |
| AG-only | 1.06 | 2.37 | 11.4 | 6.84 |
| MoG-PG | 0.71 | 2.22 | 16.9 | 4.59 |
| MoG-HG | **0.68** | **2.20** | **17.2** | **4.06** |

efficiency. Objective metrics include Fréchet Audio Distance (FAD) (Kilgour et al., 2019), Kullback-Leibler (KL) divergence (Kullback & Leibler, 1951), Inception Score (IS) (Salimans et al., 2016), Fréchet Distance (FD) (Heusel et al., 2017), and LAION-CLAP score (Wu et al., 2023). For subjective evaluation, we recruited 20 human raters to score two aspects: (i) overall perceptual quality (OVL), and (ii) semantic relevance to the input text (REL). Both scores are rated on a 1–5 scale. More details are introduced in Appendix F.5 and F.6, respectively.

## 4.2 MAIN RESULTS

**T2A generation results.** We conduct a comparison study of audio generation quality across GT (*i.e.*, ground-truth audio) and a range of systems, including AudioGen, AudioGen-Large (Kreuk et al., 2022), Make-An-Audio (Huang et al., 2023b), TANGO-AF&AC-FT-AC (Kong et al., 2024), AudioLDM-Large-Full (Liu et al., 2023), AudioLDM 2, AudioLDM 2-Large (Liu et al., 2024a), and Stable Audio Open (Evans et al., 2025). The descriptions of these models are further detailed in the Appendix F.2. For AudioGen and AudioLDM, we report the metrics as presented in their original papers, and for the rest of the methods, we cite the results from ETTA (Lee et al., 2025). To demonstrate the effectiveness of our method, we further evaluate the base model using only CFG. The best results are achieved when the CFG scale is set to $w = 7$. For a fair comparison, we fix the number of function evaluations (NFE) to 400 [2] for both our method and the CFG-only baseline, *ensuring equal inference cost*. All evaluations are conducted on the AudioCaps test set using standard objective metrics for quantitative comparison. The main results are summarized in Table 1. We have the following conclusions:

*Under equal inference resources, our method uniformly outperforms CFG-only.* In terms of audio quality, our proposed methods demonstrate significantly improved performance compared to CFG-only under the same inference budget. Specifically, both PG and HG outperform CFG-only across all objective metrics. Our results show that improved quality can be achieved *without* increasing inference cost. Notably, our HG-improved variant gives an updated result, achieving a FAD of 1.38, KL of 1.44, IS of 13.58, and a CLAP score of 0.54.

---

[2]This corresponds to 200 sampling steps for CFG and 100 for HG. For PG, we instead keep the number of sampling steps consistent with HG, yielding an NFE of 300, as we observed that further increasing it leads to slight performance degradation for PG.

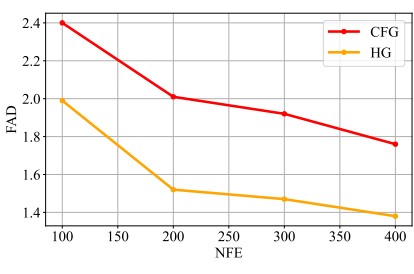 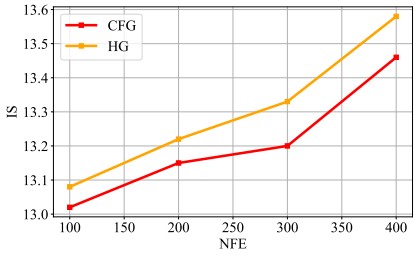

(a) FAD (↓) under different NFEs.  (b) IS (↑) under different NFEs.

Figure 3: **Performance comparison of HG and CFG-only across different NFEs.** As shown, HG consistently outperforms CFG-only across all settings.

*HG is better than PG.* Furthermore, we observe that HG consistently outperforms PG. While both methods significantly surpass the CFG-only baseline, HG achieves better performance than PG across all evaluated metrics. For example, HG achieves a lower FAD and KL than PG, indicating better distributional fidelity. Moreover, the higher CLAP score indicates that HG achieves better text-audio alignment than PG. We hypothesize these improvements stem from the structured formulation of HG, which avoids the potential conflicts present in PG.

**V2A generation results.** To further investigate the potential of AudioMoG, we also fine-tune the T2A model to V2A with CLIP features (Radford et al., 2021) and validate it on the VGGSound (Chen et al., 2020) test set. We evaluated our models using several metrics to assess audio quality, video-audio semantic alignment and temporal alignment, including FAD, KL, IS, FD, ImageBind Score (IBS) (Girdhar et al., 2023) and temporal alignment accuracy (Align acc) introduced in Diff-Foley (Luo et al., 2023). Then we compare our results with GT and a variety of strong systems, including IM2WAV (Sheffer & Adi, 2023), Diff-Foley (Luo et al., 2023), FoleyGen (Mei et al., 2024), VTA-LDM (Xu et al., 2024), FoleyCrafter (Zhang et al., 2024), V2A-Mapper (Wang et al., 2024a), VAB-Encodec (Su et al., 2024), and VATT (Liu et al., 2024b). The comparative results are summarized in Table 3. A comprehensive improvement across most metrics, including the critical issue of *temporal alignment* in V2A, demonstrates that our method uniformly enhances cross-modal audio generation. More details about the V2A experiment are provided in Appendix G.

**Text-to-music and image generation results.** To further demonstrate the effectiveness of our proposed method, we also perform our approach on text-to-music generation with our DiT base model, and conditional image generation on the public EDM (Karras et al., 2024b) checkpoint. The results are shown in Appendix C and D respectively, which again verifies the efficacy of MoG across different tasks and modalities.

**Diverse samplers.** In audio generation tasks, we use DPM++ solver, while in image generation, we use the Heun sampler, demonstrating the robustness of our methods on different samplers.

### 4.3 ADDITIONAL RESULTS

**Comparison across various NFEs.** We compare the performance of HG and CFG-only under different numbers of function evaluations (NFE), each using their respective optimal settings. Specifically, we examine NFE values of 100, 200, 300 and 400, and report the results in Figure 3. Across all NFE levels, HG consistently outperforms CFG-only in terms of FAD and IS. This consistent superiority indicates that our method is more robust across varying computational budgets, maintaining high-quality generation even under stricter inference constraints. These results highlight the strong adaptability and scalability of HG, which delivers reliable quality improvements regardless of available inference resources, while also unlocking higher performance ceilings when additional computation is allowed.

**Comparison in the same guidance scale.** To validate the effectiveness of our approach, we compare different guidance methods on both T2A and V2A. For NFE, we fix it to 300 for PG and 400 for the rest. For the guidance scale, we set $w_1 = w_3 = 1$, $w_3 = 1$, $w_1 = w_2 = 1$ and $w_2 = 1$ in the HG setting, which corresponds to no guidance, CFG-only, AG-only and PG, respectively. The remaining guidance scales are consistent with HG in Table 1 and Table 3 (AG-only also achieves the best

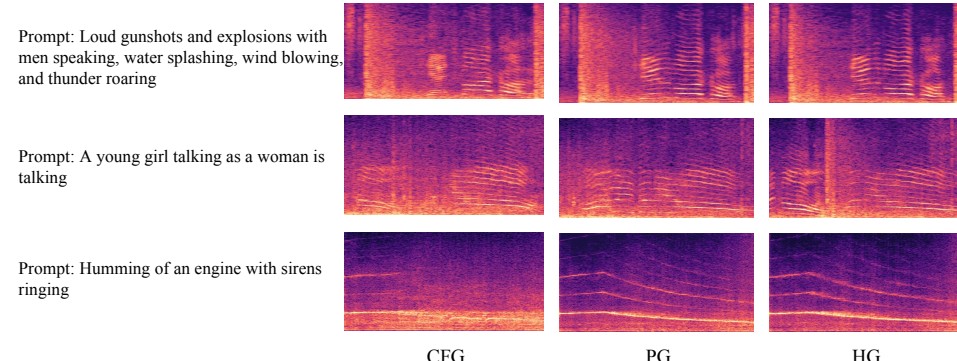

Prompt: Loud gunshots and explosions with men speaking, water splashing, wind blowing, and thunder roaring

Prompt: A young girl talking as a woman is talking

Prompt: Humming of an engine with sirens ringing

|  CFG | PG | HG |

Figure 4: **Case study comparing the spectrogram outputs of different guidance strategies (CFG, PG, HG) under various text prompts.** HG consistently demonstrates superior harmonic structure modeling and clearer spectral patterns compared to PG and CFG. While PG shows moderate improvements, CFG often struggles to capture harmonics and yields blurrier, less structured results, particularly for complex prompts. These examples visually highlight the effectiveness of hierarchical guidance in improving fidelity and temporal structure.

performance at this scale). The results are shown in Table 4 and Table 5. We observe that HG and PG outperform no guidance, CFG-only and AG-only across FAD, KL and IS, indicating that combining guidance directions yields better performance than using either one alone. Furthermore, HG surpasses PG on both tasks, further confirming the advantage of the hierarchical guidance structure.

**Impact of guidance scales.** We further investigate the influence of guidance scales in PG and HG on the audio generation quality. Our analysis reveals that these parameters have an effect on the balance between fidelity and diversity in the generated audio. Detailed experimental results and discussions are provided in Appendix B.

**Case study.** Apart from the objective and subjective evaluations, we conduct a case study as shown in Figure 4. For each of the guidance methods, CFG, PG, and HG, we provide three test text prompts. As shown, CFG is prone to produce less structured results. In comparison, PG has shown improved quality and HG produces the strongest outcome. These results are consistent with our objective test results and the human evaluations. We provided more generation results in Appendix I.

## 5 RELATED WORK

Diffusion-based cross-modal audio generation, such as T2A and V2A generation systems, primarily employ CFG during sampling, while it may suffer from sub-optimal synthesis quality as discussed. Recent work (Chidambaram et al., 2024) has theoretically justified the empirical finding that a large guidance scale which can ensure synthesis quality may result in reduced generation diversity. To address this issue, recent works (Karras et al., 2024a; Hong et al., 2023; Li et al., 2025; Sadat et al., 2024) carefully design different weaker models to replace the unconditional term in CFG, as noted in Jeon (2025). Among them, AG is representative as its general formulation, and has been extended to diverse data modalities and generation tasks (Phunyaphibarn et al., 2025; Jeon, 2025; Hyung et al., 2025; Lee et al., 2025). In this work, we propose a mixture-of-guidance framework, exploiting complementary advantages of different guidance methods, aiming at improving guided audio generation quality without sacrificing inference efficiency.

## 6 CONCLUSION

In this work, we introduce AudioMoG, an improved sampling framework for audio generation. We start from an analysis of CFG and AG methods, demonstrating their respective principle on outperforming unguided diffusion generation. Then, we propose the mixture-of-guidance framework, introducing its mechanism which exploits the *cumulative benefits* of diverse guidance methods. Comprehensive experiments demonstrate the superiority of AudioMoG over the popularly adopted CFG on T2A generation under the same inference budgets, as well as achieving improvement over both CFG and AG across V2A, T2M, and image generation tasks without increasing inference time.

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

CONTENTS

# A    PROOF OF HG

**Theorem 1.** *Embedding AG into CFG is equivalent to embedding CFG into AG. Namely,*

$$\boldsymbol{\epsilon}_{cAG}(\boldsymbol{z}_t, t, \boldsymbol{c}) = \boldsymbol{\epsilon}_{\theta_{bad}}(\boldsymbol{z}_t, t, \boldsymbol{c}) + w_1(\boldsymbol{\epsilon}_\theta(\boldsymbol{z}_t, t, \boldsymbol{c}) - \boldsymbol{\epsilon}_{\theta_{bad}}(\boldsymbol{z}_t, t, \boldsymbol{c}))$$
$$\boldsymbol{\epsilon}_{ucAG}(\boldsymbol{z}_t, t) = \boldsymbol{\epsilon}_{\theta_{bad}}(\boldsymbol{z}_t, t) + w_2(\boldsymbol{\epsilon}_\theta(\boldsymbol{z}_t, t) - \boldsymbol{\epsilon}_{\theta_{bad}}(\boldsymbol{z}_t, t)) \tag{11}$$
$$\boldsymbol{\epsilon}_{HG}(\boldsymbol{z}_t, t, \boldsymbol{c}) = \boldsymbol{\epsilon}_{ucAG}(\boldsymbol{z}_t, t) + w_3(\boldsymbol{\epsilon}_{cAG}(\boldsymbol{z}_t, t, \boldsymbol{c}) - \boldsymbol{\epsilon}_{ucAG}(\boldsymbol{z}_t, t))$$

*and*

$$\boldsymbol{\epsilon}_{CFG}(\boldsymbol{z}_t, t, \boldsymbol{c}) = \boldsymbol{\epsilon}_\theta(\boldsymbol{z}_t, t) + \hat{w}_1(\boldsymbol{\epsilon}_\theta(\boldsymbol{z}_t, t, \boldsymbol{c}) - \boldsymbol{\epsilon}_\theta(\boldsymbol{z}_t, t))$$
$$\boldsymbol{\epsilon}_{badCFG}(\boldsymbol{z}_t, t, \boldsymbol{c}) = \boldsymbol{\epsilon}_{\theta_{bad}}(\boldsymbol{z}_t, t) + \hat{w}_2(\boldsymbol{\epsilon}_{\theta_{bad}}(\boldsymbol{z}_t, t, \boldsymbol{c}) - \boldsymbol{\epsilon}_{\theta_{bad}}(\boldsymbol{z}_t, t)) \tag{12}$$
$$\hat{\boldsymbol{\epsilon}}(\boldsymbol{z}_t, t, \boldsymbol{c}) = \boldsymbol{\epsilon}_{badCFG}(\boldsymbol{z}_t, t, \boldsymbol{c}) + \hat{w}_3(\boldsymbol{\epsilon}_{CFG}(\boldsymbol{z}_t, t, \boldsymbol{c}) - \boldsymbol{\epsilon}_{badCFG}(\boldsymbol{z}_t, t, \boldsymbol{c}))$$

*are equivalent guidance, as long as $w_3 \notin \{0, 1\}$ and $\hat{w}_3 \notin \{0, 1\}$ (HG degenerates to CFG-only or AG-only in these cases) i.e., for any fixed $\boldsymbol{\epsilon}_\theta$ and $\boldsymbol{\epsilon}_{\theta_{bad}}$ ,*

$$\{\boldsymbol{\epsilon} | \exists w_1, w_2 \in \mathbb{R}, w_3 \notin \{0, 1\}, s.t. \boldsymbol{\epsilon} = \boldsymbol{\epsilon}_{HG}(\boldsymbol{z}_t, t, \boldsymbol{c})\}$$
$$= \{\boldsymbol{\epsilon} | \exists \hat{w}_1, \hat{w}_2 \in \mathbb{R}, \hat{w}_3 \notin \{0, 1\}, s.t. \boldsymbol{\epsilon} = \hat{\boldsymbol{\epsilon}}_{HG}(\boldsymbol{z}_t, t, \boldsymbol{c})\} \tag{13}$$

**Proof**. Let the two sets in Equation 13 be denoted as $S_1$ and $S_2$. Substituting the first two equations of Equation 11 into the third gives:

$$\boldsymbol{\epsilon}_{\mathrm{HG}}(\boldsymbol{z}_t, t, \boldsymbol{c}) = w_3 \boldsymbol{\epsilon}_{\mathrm{cAG}}(\boldsymbol{z}_t, t, \boldsymbol{c}) + (1 - w_3)\boldsymbol{\epsilon}_{\mathrm{ucAG}}(\boldsymbol{z}_t, t, \boldsymbol{c})$$
$$= w_3 \left( w_1 \boldsymbol{\epsilon}_\theta(\boldsymbol{z}_t, t, \boldsymbol{c}) + (1 - w_1)\boldsymbol{\epsilon}_{\theta_{\mathrm{bad}}}(\boldsymbol{z}_t, t, \boldsymbol{c}) \right)$$
$$+ (1 - w_3) \left( w_2 \boldsymbol{\epsilon}_\theta(\boldsymbol{z}_t, t) + (1 - w_2)\boldsymbol{\epsilon}_{\theta_{\mathrm{bad}}}(\boldsymbol{z}_t, t) \right) \tag{14}$$
$$= w_1 w_3 \boldsymbol{\epsilon}_\theta(\boldsymbol{z}_t, t, \boldsymbol{c}) + w_2(1 - w_3)\boldsymbol{\epsilon}_\theta(\boldsymbol{z}_t, t)$$
$$+ w_3(1 - w_1)\boldsymbol{\epsilon}_{\theta_{\mathrm{bad}}}(\boldsymbol{z}_t, t, \boldsymbol{c}) + (1 - w_2)(1 - w_3)\boldsymbol{\epsilon}_{\theta_{\mathrm{bad}}}(\boldsymbol{z}_t, t)$$

Likewise, Equation 12 gives

$$\hat{\boldsymbol{\epsilon}}_{\mathrm{HG}}(\boldsymbol{z}_t, t, \boldsymbol{c}) = \hat{w}_3 \boldsymbol{\epsilon}_{\mathrm{CFG}}(\boldsymbol{z}_t, t, \boldsymbol{c}) + (1 - \hat{w}_3)\boldsymbol{\epsilon}_{\mathrm{badCFG}}(\boldsymbol{z}_t, t, \boldsymbol{c})$$
$$= \hat{w}_3 \left( \hat{w}_1 \boldsymbol{\epsilon}_\theta(\boldsymbol{z}_t, t, \boldsymbol{c}) + (1 - \hat{w}_1)\boldsymbol{\epsilon}_\theta(\boldsymbol{z}_t, t) \right)$$
$$+ (1 - \hat{w}_3) \left( \hat{w}_2 \boldsymbol{\epsilon}_{\theta_{\mathrm{bad}}}(\boldsymbol{z}_t, t, \boldsymbol{c}) + (1 - \hat{w}_2)\boldsymbol{\epsilon}_{\theta_{\mathrm{bad}}}(\boldsymbol{z}_t, t) \right) \tag{15}$$
$$= \hat{w}_1 \hat{w}_3 \boldsymbol{\epsilon}_\theta(\boldsymbol{z}_t, t, \boldsymbol{c}) + \hat{w}_3(1 - \hat{w}_1)\boldsymbol{\epsilon}_\theta(\boldsymbol{z}_t, t)$$
$$+ \hat{w}_2(1 - \hat{w}_3)\boldsymbol{\epsilon}_{\theta_{\mathrm{bad}}}(\boldsymbol{z}_t, t, \boldsymbol{c}) + (1 - \hat{w}_2)(1 - \hat{w}_3)\boldsymbol{\epsilon}_{\theta_{\mathrm{bad}}}(\boldsymbol{z}_t, t)$$

For most cases $\boldsymbol{\epsilon}_\theta(\boldsymbol{z}_t, t)$, $\boldsymbol{\epsilon}_\theta(\boldsymbol{z}_t, t, \boldsymbol{c})$, $\boldsymbol{\epsilon}_{\theta_{\mathrm{bad}}}(\boldsymbol{z}_t, t)$ and $\boldsymbol{\epsilon}_{\theta_{\mathrm{bad}}}(\boldsymbol{z}_t, t, \boldsymbol{c})$ are linearly independent, thus Equation 14 and 15 implies

$$\begin{cases} w_1 w_3 = \hat{w}_1 \hat{w}_3 \\ w_2(1 - w_3) = \hat{w}_3(1 - \hat{w}_1) \\ w_3(1 - w_1) = \hat{w}_2(1 - \hat{w}_3) \\ (1 - w_2)(1 - w_3) = (1 - \hat{w}_2)(1 - \hat{w}_3) \end{cases} \tag{16}$$

For Equations 16, adding the first and third equations yields

$$w_3 = \hat{w}_1 \hat{w}_3 + \hat{w}_2(1 - \hat{w}_3) \tag{17}$$

Substitute into the first and second equations and we can figure out (This division is valid since $w_3 \notin \{0, 1\}$)

$$w_1 = \frac{\hat{w}_1 \hat{w}_3}{\hat{w}_1 \hat{w}_3 + \hat{w}_2(1 - \hat{w}_3)}$$
$$w_2 = \frac{\hat{w}_3(1 - \hat{w}_1)}{1 - \hat{w}_1 \hat{w}_3 - \hat{w}_2(1 - \hat{w}_3)} \tag{18}$$

Therefore, $S_2 \subseteq S_1$. Notice that Equations 16 is symmetric under the exchange of $w_1$ and $\hat{w}_1$, $w_2$ and $\hat{w}_2$, $w_3$ and $\hat{w}_3$, so similarly

$$\hat{w}_1 = \frac{w_1 w_3}{w_1 w_3 + w_2(1 - w_3)}$$
$$\hat{w}_2 = \frac{w_3(1 - w_1)}{1 - w_1 w_3 - w_2(1 - w_3)} \tag{19}$$
$$\hat{w}_3 = w_1 w_3 + w_2(1 - w_3)$$

deducing that $S_1 \subseteq S_2$. Then we have $S_1 = S_2$, which completes the proof.

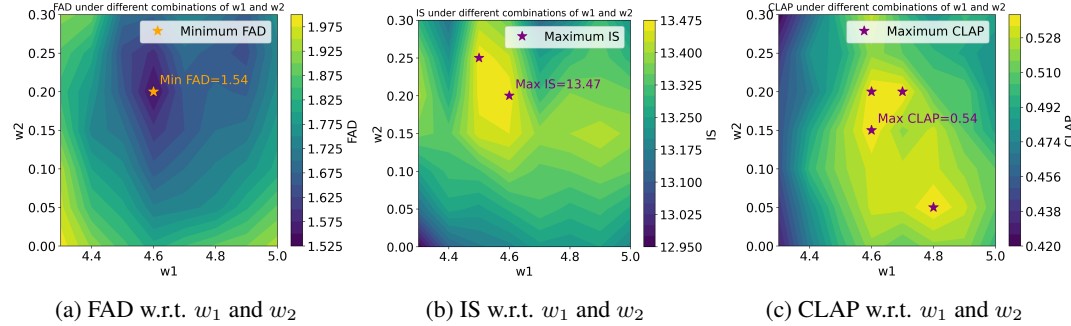

(a) FAD w.r.t. $w_1$ and $w_2$      (b) IS w.r.t. $w_1$ and $w_2$      (c) CLAP w.r.t. $w_1$ and $w_2$

Figure 5: **Impact of different guidance scales in PG. (a)(b)(c)** stands for the relations of FAD, IS and CLAP with $w_1$ and $w_2$, respectively. The best configurations are denoted with stars.

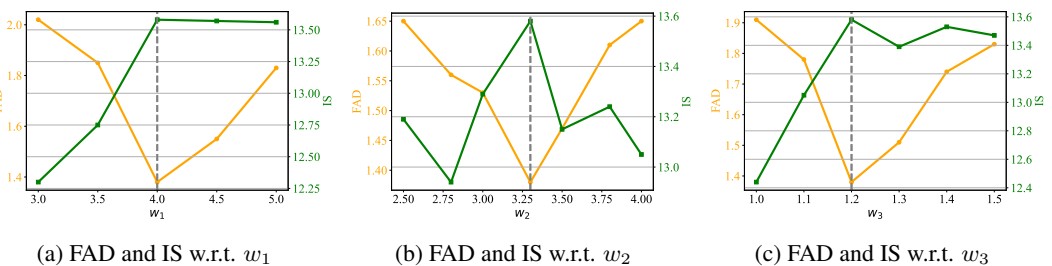

(a) FAD and IS w.r.t. $w_1$      (b) FAD and IS w.r.t. $w_2$      (c) FAD and IS w.r.t. $w_3$

Figure 6: **Impact of different guidance scales in HG. (a)** Sweep over $w_1$ while keeping $w_2$ and $w_3$ unchanged. **(b)** Sweep over $w_2$ while keeping $w_1$ and $w_3$ unchanged. **(c)** Sweep over $w_3$ while keeping $w_1$ and $w_2$ unchanged. The best configurations are denoted with dashed lines.

## B   IMPACT OF GUIDANCE SCALES

To further demonstrate the effectiveness of AudioMoG, we investigate the impact of different guidance scales in PG and HG on the generation results.

**The PG setting.** We study the guidance scales $w_1$ in the range $4.3 \sim 5.0$ and $w_2$ in $0.0 \sim 0.3$. We keep the NFE fixed at 300 following the main paper and evaluated FAD, IS and CLAP. The results are shown in Figures 5a-5c. Noticing that PG degrades to CFG-only when $w_2 = 0$, we can see that PG indeed reaches a higher performance ceiling and achieves the best results around our choice, $w_1 = 4.6$ and $w_2 = 0.2$. Excessively increasing or decreasing $w_1$ and $w_2$ will result in a loss of quality, showing alike convex pattern in the following HG setting.

**The HG setting.** We adopt the setting in the main paper, with the baseline configuration of $w_1 = 4.0$, $w_2 = 3.3$, and $w_3 = 1.2$, while keeping the NFE fixed at 400. We evaluated both FAD and IS as primary metrics, and the results are summarized in Figures 6a-6c. For $w_1$ (Figure 6a), we can see as $w_1$ increases from 3.0 to 4.0, FAD decreases substantially, reaching a minimum around $w_1 = 4.0$, indicating improved overall sample fidelity. However, further increasing $w_1$ beyond this point slightly worsens FAD. Meanwhile, IS increases steadily and saturates at $w_1 \geq 4.0$, reflecting stronger condition adherence and sample specificity. For $w_2$ and $w_3$, we observe similar convex patterns (Figure 6b, 6c). Both FAD and IS achieve their best values around $w_2 = 3.3$ and $w_3 = 1.2$. Therefore, all three scales exhibit non-monotonic behaviors, and each has an optimal value range where both fidelity and diversity metrics are simultaneously optimized. These results validate the necessity of tuning each guidance component, as also noted in Karras et al. (2024a), and highlight that balanced guidance from multiple perspectives is crucial for high-quality audio generation.

## C   TEXT-TO-MUSIC GENERATION RESULTS

### C.1   BASELINE METHODS

To present comprehensive evaluation results, we introduce 10 text-to-music (T2M) baseline models for comparison:

Table 6: **Objective metrics for text-to-music generation on MusicCaps dataset.**

| Model | FAD ↓ | IS ↑ | FD ↓ |
|---|---|---|---|
| Riffusion (Forsgren & Martiros, 2022) | 13.40 | / | / |
| Mubert (Mubert-Inc., 2022) | 9.60 | / | / |
| MusicLM (Agostinelli et al., 2023) | 4.00 | / | / |
| Mousai (Schneider et al., 2023) | 7.50 | / | / |
| MeLoDy (Lam et al., 2023) | 5.41 | / | / |
| Stable Audio Open (Evans et al., 2025) | 3.51 | 2.93 | 36.42 |
| MusicGen w/o melody (Copet et al., 2023) | 3.40 | / | / |
| AudioLDM 2-Large (Liu et al., 2024a) | 2.93 | 2.59 | 16.34 |
| AudioLDM 2-Full (Liu et al., 2024a) | 3.13 | / | / |
| TANGO-AF (Kong et al., 2024) | 2.21 | 2.79 | 22.69 |
| Jen-1 (Li et al., 2024b) | 2.00 | / | / |
| CFG-only, $w = 7$ | 2.36 | 3.10 | 14.39 |
| MoG-PG, $w_1 = 2.0, w_2 = 0.2$ | 1.98 | 4.35 | 14.88 |
| MoG-HG, $w_1 = 1.6, w_2 = 1.6, w_3 = 1.2$ | **1.92** | **4.39** | **14.09** |

**Riffusion** (Forsgren & Martiros, 2022) is a unique model that generates music by converting spectrogram images into audio. It fine-tunes the Stable Diffusion model on spectrograms, allowing it to produce short music loops based on text prompts.

**Mubert** (Mubert-Inc., 2022) is an AI-driven music generation platform that creates royalty-free music tailored for various content needs. It offers tools for content creators, artists, and developers to generate and integrate AI-generated music into their projects.

**MusicLM** (Agostinelli et al., 2023) is a model introduced by Google that generates high-fidelity music from text descriptions. It utilizes a sequence-to-sequence modeling approach to capture long-term structure in music generation.

**Mousai** (Schneider et al., 2023) is a two-stage latent-diffusion system for text-to-music generation. Stage 1 compresses 48 kHz stereo audio with a Diffusion-Magnitude Autoencoder (DMAE), and Stage 2 is a text-conditioned latent diffusion (TCLD) model that can produce multi-minute, prompt-aligned musical pieces.

**MeLoDy** (Lam et al., 2023) is an LM-guided diffusion framework that inherits the highest-level LM from MusicLM for semantic modeling, and applies a novel dual-path diffusion (DPD) model and an audio VAE-GAN to efficiently decode the conditioning semantic tokens into waveform, resulting in cutting sampling cost by more than 95 % while maintaining state-of-the-art text–music alignment and audio quality.

**MusicGen** (Copet et al., 2023) is an open-source model developed by Meta that generates music from text prompts. It employs a transformer-based architecture trained on a large dataset of music to produce diverse and high-quality audio samples.

**Jen-1** (Li et al., 2024b) is a universal high-fidelity model for text-to-music generation. It incorporates both autoregressive and non-autoregressive training, enabling tasks like text-guided music generation, inpainting, and continuation.

## C.2 EXPERIMENT RESULTS

We conduct a comprehensive analysis of generated music quality across the above systems. For Riffusion, Mubert and MusicLM, we report the metrics from MusicLM. For Stable Audio Open, AudioLDM2-Large and Tango-AF, we cite the results from ETTA. For Mousai, as it is not evaluated in ETTA, we report it from AudioLDM 2. For other baselines, we report the metrics as presented in their original papers. Similar to T2A, we further evaluate our baseline model using only CFG, which also achieves the best result when $w = 7$, and fix NFE to 400. All evaluations are conducted on MusicCaps dataset. The results are detailed in Table 6. It can be shown that HG not only surpasses CFG-only but also achieves SOTA in all metrics.

# D    IMAGE GENERATION RESULTS

This section presents the experimental results for the conditional ImageNet $512 \times 512$ generation task, using the EDM2-S checkpoints from (Karras et al., 2024b). Image generation was performed over 16 deterministic steps with a second-order Heun sampler, maintaining the same inference speed with a single guidance method (Karras et al., 2024a). The optimal guidance strengths were determined through a grid search on a reduced sample size ($N = 8192$). Subsequently, the model's performance was evaluated on a larger set of samples ($N = 50000$) to obtain robust estimates of the Fréchet Inception Distance (FID) and the Fréchet DINOv2 Distance (FD_DINOv2).

Table 7: **Conditional image generation results on ImageNet-512.**

| **Model** | **FID** $\downarrow$ | **FD**$_{\text{DINOv2}}$ $\downarrow$ |
|---|---|---|
| CFG-only | 2.40 | 96.90 |
| AG-only | 1.60 | 57.35 |
| MoG-PG | 1.60 | 53.01 |
| MoG-HG | **1.47** | **49.92** |

# E    2D TOY DATASET EXPERIMENT DETAILS

This section outlines the setup of the 2D toy dataset experiment used in the analysis presented in Section 3.1. Unless otherwise specified, the experiment details strictly follow the setup in (Karras et al., 2024a).

**Dataset.** The dataset is a synthetic, fractal-like 2D distribution composed of two classes. Each class is represented as a Gaussian mixture model $\mathcal{M}_c = (\phi_i, \mu_i, \Sigma_i)$, where $\phi_i$ denotes the mixture weight, $\mu_i$ the mean, and $\Sigma_i$ the $2 \times 2$ covariance matrix of the $i$-th Gaussian component.

**Models.** We employ simple multi-layer perceptrons (MLPs) as denoiser models, consistent with the setup in (Karras et al., 2024a).

**Training.** To ensure comparability, we use the pre-trained models provided by (Karras et al., 2024a). The URLs of model checkpoints can be found at: `https://github.com/NVlabs/edm2/blob/main/toy_example.py`.

**Sampling.** For the visualizations in Figure 2, we use the following guidance weights: $w = 3$ for both CFG and AG; $w_1 = 2, w_2 = 2$ for PG; and $w_1 = w_2 = 1.5$, $w_3 = 2$ for HG. The models $\epsilon_\theta(z_t, t, c)$ and $\epsilon_\theta(z_t, t)$ use checkpoints trained with hidden dimension $d = 64$ and training iteration $M = 4096$, while the bad models $\epsilon_{\theta_{\text{bad}}}(z_t, t, c)$ and $\epsilon_{\theta_{\text{bad}}}(z_t, t)$ use checkpoints with $d = 32$ and $M = 512$.

# F    TEXT-TO-AUDIO EXPERIMENT DETAILS

## F.1    DATASETS

We present the datasets used to train our baseline model in Table 8. AudioCaps (Kim et al., 2019) is a benchmark dataset for audio captioning that contains 50,000 audio clips from AudioSet paired with human-written textual descriptions. We only used its training set. AudioSet (Gemmeke et al., 2017) is a large-scale weakly labeled dataset released by Google, comprising over 2 million 10-second audio clips across more than 600 sound event categories. The BBC Sound Effects Library [link] provides over 30,000 professionally recorded sound effects covering a wide variety of acoustic scenes and events. Clotho v2 (Drossos et al., 2020) is designed for audio captioning tasks and contains approximately 5,000 audio clips, each annotated with five crowdsourced textual descriptions. VGGSound (Chen et al., 2020) is a large-scale dataset consisting of over 200,000 10-second video clips from YouTube, covering 310 diverse sound classes. FreeSound [link] is a collaborative platform that hosts a wide range of user-contributed audio samples, frequently used for environmental sound classification and retrieval. FSD50K (Fonseca et al., 2021) is a large-scale dataset derived from FreeSound, containing over 50,000 audio clips annotated with strong and weak labels for sound

event detection. FMA (Defferrard et al., 2016), the Free Music Archive dataset, includes full-length high-quality music tracks and is widely used in music information retrieval research. The Million Song Dataset (MSD) (Bertin-Mahieux et al., 2011) offers metadata and pre-computed audio features for one million popular music tracks to support large-scale music recommendation and analysis. MagnaTagATune (MTT) (Law et al., 2009) is a music tagging dataset that contains 25,000 audio clips annotated with multiple descriptive tags for genre, instrument, and mood classification tasks.

Table 8: **Statistics for the datasets used in the paper.**

| **Dataset** | **Hours** (h) | **Source** |
|---|---|---|
| AudioCaps | 109 | (Kim et al., 2019) |
| AudioSet | 5800 | (Gemmeke et al., 2017) |
| BBC Sound Effects Library | 300 | link |
| Clotho v2 | 152 | (Drossos et al., 2020) |
| VGGSound | 550 | (Chen et al., 2020) |
| FreeSound | 6246 | link |
| FSD50k | 108 | (Fonseca et al., 2021) |
| FMA | 900 | (Defferrard et al., 2016) |
| MSD | 7333 | (Bertin-Mahieux et al., 2011) |
| MTT | 200 | (Law et al., 2009) |

### F.2 BASELINE METHODS

We employ 6 strong T2A baseline methods for comparison.

**AudioLDM** is a latent diffusion model for text-to-audio (T2A) generation presented by (Liu et al., 2023). It performs the diffusion process in the latent space of a pretrained audio VAE, while conditioning on text embeddings produced by the CLAP text branch. This design enables a T2A training process without paired data and produces audio that is semantically consistent with the input description.

**AudioLDM2** is an improved version of the AudioLDM model, introduced by (Liu et al., 2024a). It incorporates several improvements, including leveraging the Language of Audio (LOA) encoder and finetuning a GPT-2 model to translate any modality to LOA. These improvements result in higher-quality audio generation that better aligns with the input text.

**AudioGen** is a text-to-audio generation model introduced by (Kreuk et al., 2022). It employs an autoregressive transformer architecture to generate audio samples conditioned on textual descriptions. The model is trained on a large-scale dataset of audio-text pairs, enabling it to produce high-quality audio outputs that align with the given textual input.

**Make-An-Audio** is a diffusion-based text-to-audio generation model proposed by (Huang et al., 2023b). It introduces a pseudo prompt enhancement with the distill-then-reprogram approach, including a large number of concept compositions by opening up the usage of language-free audios to alleviate data scarcity. Therefore, it enables high-fidelity, prompt-aligned outputs.

**TANGO-AF&AC-FT-AC** (Kong et al., 2024) pre-trains the TANGO architecture on the synthetic-caption AF-AudioSet plus AudioCaps, followed by fine-tuning on AudioCaps alone. Leveraging high-quality synthetic captions significantly improves text-to-audio alignment and overall audio realism.

**Stable Audio Open** (Evans et al., 2025) is an open-source text-to-audio generation model developed by Stability AI. It leverages a diffusion-based architecture trained on a diverse dataset of audio-text pairs. The model is designed to generate high-fidelity audio samples conditioned on textual input, supporting various applications such as music generation, sound effect synthesis, and more.

### F.3 MODEL CONFIGURATIONS

Our diffusion model is built upon the Diffusion Transformer (DiT) (Peebles & Xie, 2023) architecture, following a latent diffusion modeling (LDM) (Rombach et al., 2022) paradigm that offers strong

generative capabilities and effective context modeling. The backbone of the diffusion network adopts a DiT structure with 24 layers and 24 attention heads, each with an embedding dimension of 1536. The model supports both cross-attention and global conditioning: cross-attention is applied to all types of conditional inputs, while global conditioning specifically handles duration-related control signals. The internal token dimension of the diffusion model is set to 64, with a conditional token dimension of 768 and a global condition embedding dimension of 1536. The generated latent representation has the same dimensionality as `io_channels`, which is 64.

### F.4 COMPRESSION NETWORKS

To train the audio autoencoder, we adopt a variational autoencoder (VAE) (Kingma et al., 2013) architecture based on the Oobleck framework (Evans et al., 2025) at a sampling rate of 16kHz. The model is trained from scratch on large-scale publicly available text-audio paired datasets. The encoder and decoder are symmetric, each using a base channel size of 128, with channel multipliers $1, 2, 4, 8, 16$ and strides $2, 2, 4, 4, 10$. The encoder maps the input waveform into a 128-dimensional latent representation, while the decoder reconstructs the waveform from a 64-dimensional latent code. Snake activation is applied throughout the network, and no final tanh activation is used in the decoder. The overall downsampling ratio is 640, and both input and output are mono-channel waveforms. The bottleneck is implemented as a variational layer.

### F.5 OBJECTIVE METRICS

We introduce the objective metrics employed in our evaluation, including Fréchet Audio Distance (FAD) (Kilgour et al., 2019), Kullback-Leibler (KL) divergence (Kullback & Leibler, 1951), Inception Score (IS) (Salimans et al., 2016), Fréchet Distance (FD) (Heusel et al., 2017), and LAION-CLAP score (Wu et al., 2023). FAD, adapted from FID (Heusel et al., 2017), measures the distributional gap between generated and reference audio using VGGish embeddings (Hershey et al., 2017), and serves as our primary indicator of audio fidelity. KL divergence evaluates the difference in acoustic event posteriors between ground truth and generated audio, computed using the PANN tagging model (Kong et al., 2020). IS reflects both diversity and specificity of the generated samples, based on entropy over class predictions. FD, while similar in formulation to FAD, is computed in more general embedding spaces and tends to be less stable in audio tasks. We include it for completeness but primarily rely on FAD for fidelity assessment. The CLAP score is calculated as the cosine similarity between CLAP embeddings of the generated audio and the corresponding text. We use the AudioLDM evaluation toolkit to compute all objective metrics.

### F.6 SUBJECTIVE EVALUATION

We randomly selected 20 samples from the AudioCaps test set for the subjective evaluation. Each group includes the results from AudioLDM, AudioLDM2, CFG-only, HG, and the ground truth (GT), with the order of samples within each group randomly shuffled. Each group was rated by 20 human raters. In our evaluation, both overall quality (OVL) and text relevance (REL) are rated on a scale from 1 to 5. For OVL, raters assess the perceptual quality of the audio, while for REL, they rate the relevance of the audio to the given text condition. The minimum rating increment for all scores is 1 point. A screenshot of our evaluation interface is shown in Figure 7.

## G VIDEO-TO-AUDIO EXPERIMENT DETAILS

### G.1 DATASETS

Apart from T2A and T2M generation, we conduct experiments for video-to-audio(V2A) generation. We utilize the benchmark datasets AudioSet (Gemmeke et al., 2017) and VGGSound (Chen et al., 2020) for model training. To compare with previous work, we evaluated our models on the VGGSound test set, which consists of about 15K 10-second audio clips.

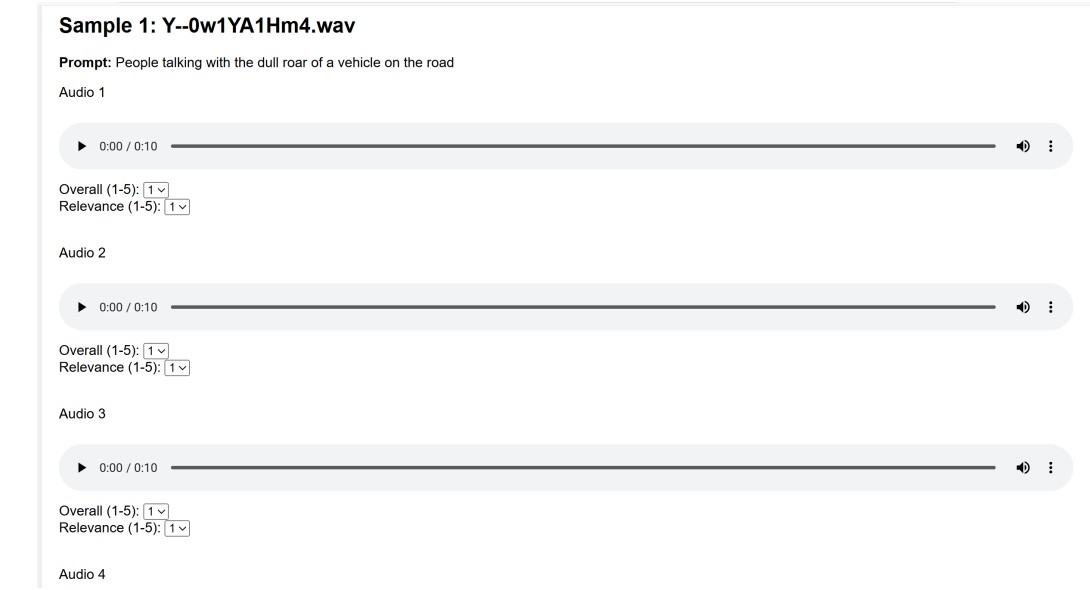

Figure 7: **Screenshot of our subjective evaluations.**

## G.2 MODEL CONFIGURATIONS

We use CLIP (Radford et al., 2021) embeddings to extract the visual features, and we leverage the same VAE as in T2A. The diffusion backbone shares the same core architecture as the T2A counterpart, being built upon the DiT within the LDM paradigm. Most hyperparameters remain the same with T2A, but we increased the conditional token dimension to 1024 to better accommodate high-dimensional visual embeddings, and the global condition embedding dimension is set to 2048.

We fine-tuned from a T2A model that was trained for 2M iterations with a batch size of 8 per GPU. For the good model, we conducted 1.3M finetuning iterations, while for the bad model, we fine-tuned for 0.3M iterations, both using a batch size of 8 per GPU. The optimizer and sampler settings are the same as those used in the T2A model.

## G.3 BASELINES METHODS

We introduce 8 V2A baseline models for comparison.

**IM2WAV** is an image-guided open-domain audio generation system introduced by (Sheffer & Adi, 2023). It employs two Transformer language models operating over a discrete audio representation derived from a VQ-VAE (Van Den Oord et al., 2017) model. The system first generates a low-level audio representation using a language model, then upsamples the audio tokens with an additional language model to produce high-fidelity audio. Visual conditioning is achieved through CLIP embeddings, and CFG is applied to steer the generation process.

**Diff-Foley** is a synchronized V2A synthesis method utilizing a latent diffusion model (LDM), presented by (Luo et al., 2023). It incorporates contrastive audio-visual pretraining (CAVP) to learn temporally and semantically aligned features, which are then used to train the LDM on spectrogram latent space. The model employs cross-attention modules and "double guidance" to enhance sample quality, achieving state-of-the-art performance in V2A tasks.

**FoleyGen** is an open-domain V2A generation system based on a language modeling paradigm, introduced by (Mei et al., 2024). It leverages a neural audio codec for bidirectional conversion between waveforms and discrete tokens. A single Transformer model, conditioned on visual features extracted from a visual encoder, facilitates the generation of audio tokens. The model addresses temporal synchronization challenges by exploring novel visual attention mechanisms.

**VTA-LDM** is a V2A generation framework developed by (Xu et al., 2024), building upon the LDM framework. It employs a CLIP-based vision encoder to extract frame-level video features, which are temporally concatenated and mapped using a projector as the generation condition. The model focuses on generating semantically and temporally aligned audio content corresponding to video inputs.

**FoleyCrafter** (Zhang et al., 2024) is a text-based V2A generation framework designed to produce high-quality, semantically relevant, and temporally synchronized audio for videos. It extends state-of-the-art T2A generators by incorporating a semantic adapter for semantic alignment and a temporal adapter for precise audio-video synchronization, ensuring realistic sound effects that align with visual content.

**V2A-Mapper** is a lightweight solution for V2A generation proposed by (Wang et al., 2024a). It connects foundation models by translating visual CLIP embeddings into auditory CLAP embeddings, bridging the domain gap between visual and audio modalities. Conditioned on the translated CLAP embedding, a pretrained audio generative model (AudioLDM) is used to produce high-fidelity and visually-aligned sound, requiring minimal training parameters.

**VAB-Encodec** (Su et al., 2024) is a unified audio-visual framework that learns latent representations and enables vision-to-audio generation within the same model. It tokenizes 48 kHz audio with a pretrained Encodec tokenizer and encodes video frames with an image encoder. During pre-training the model performs visual-conditioned masked-audio-token prediction; at inference it iteratively decodes audio tokens conditioned on visual features, yielding fast and semantically aligned sound.

**VATT** is a multi-modal generative framework for V2A generation through text, presented by (Liu et al., 2024b). It comprises two modules: VATT Converter, a large language model fine-tuned for instructions that maps video features to the LLM vector space; and VATT Audio, a transformer that generates audio tokens from visual frames and optional text prompts using iterative parallel decoding. The framework allows for controllable audio generation and audio captioning based on video inputs.

For Diff-foley, VTA-LDM and FoleyCrafter, we generate 10-second audio samples using their official implementations. For V2A Mapper, it supplies pre-generated audio samples for evaluation. As the official implementations of FoleyGen, VAB-Encodec and VATT are unavailable, we compare our results with their official reported results. The IM2WAV results are adopted from VATT.

G.4 METRICS

We introduce additional metrics utilized in our V2A evaluation apart from FAD, KL, IS and FD, including Imagebind Score (IBS) and temporal alignment accuracy (Align acc), which primarily measures audio-visual alignment.

**ImageBind Score (IBS)** assesses the semantic alignment between generated audio and the corresponding video by computing the cosine similarity between their embeddings in a shared multimodal space. This metric leverages the ImageBind model (Girdhar et al., 2023), which aligns multiple modalities—including images, audio, and text—into a unified embedding space, facilitating cross-modal retrieval and evaluation . A higher IBS indicates a stronger semantic correlation between the audio and video content.

**Temporal Alignment Accuracy (Align Acc)** measures the synchronization between generated audio and video by evaluating the model's ability to produce audio events that are temporally aligned with visual events. Introduced in Diff-Foley (Luo et al., 2023), this metric involves training a classifier to distinguish between correctly aligned audio-video pairs and misaligned ones. The classifier is trained on three types of pairs: true pairs (correctly aligned), temporally shifted pairs, and mismatched pairs from different videos. Align Acc is computed as the percentage of correctly identified true pairs, providing a quantitative measure of temporal synchronization.

By incorporating both IBS and Align Acc, we offer a comprehensive evaluation of the semantic and temporal alignment between generated audio and video, ensuring that the audio not only matches the content but also aligns accurately in time.

# H DETAILED RELATED WORKS

## H.1 TEXT-TO-AUDIO (T2A) GENERATION

T2A systems generate audio samples conditioned on natural language prompts. At the beginning, DiffSound (Yang et al., 2023), AudioGen (Kreuk et al., 2022) explore autoregressive-based generation methods in the compressed space of mel-spectrogram and waveform respectively. Then, AudioLDM (Liu et al., 2023) and Make-An-Audio (Huang et al., 2023b) develop latent diffusion models in the compressed space of mel-spectrogram, improving overall T2A generation quality. Tango (Deepanway et al., 2023) improves the text encoder of diffusion-based T2A systems with a language model. AudioLDM 2 (Liu et al., 2024a) employs an autoregressive-based method to predict the AudioMAE (Huang et al., 2022) features from various input modalities, and then uses a latent diffusion model to generate audio from AudioMAE features. Tango2 (Majumder et al., 2024) and Tangoflux (Hung et al., 2024) utilize reinforcement strategies including direct preference optimization (Rafailov et al., 2023) and CLAP-guided reward shaping (Wu et al., 2023) to improve the human preference and semantic-textual alignment. Recently, Stable Audio (Evans et al., 2024) designs transformer-based scalable latent diffusion models in the space directly compressed from the audio waveform. ETTA (Lee et al., 2025) elucidates the design space of diffusion-based T2A systems.

These innovative methods have improved T2A generation quality from generative methods, compression networks, and network architectures, while the innovations on guidance methods have not been carefully investigated in previous works.

## H.2 VIDEO-TO-AUDIO (V2A) GENERATION

Recent advances in video-to-audio (V2A) generation can be broadly divided into two categories: (1) enhancing V2A via pre-trained text-to-audio (T2A) models, and (2) introducing auxiliary temporal representations to improve temporal alignment. In the first category, methods such as V2A-Mapper (Wang et al., 2024a) and FoleyCrafter (Mei et al., 2024) build upon established T2A systems like AudioLDM (Liu et al., 2023). These approaches either align video features with the original conditioning space of T2A models or introduce additional adapters to inject visual information as supplementary conditions. For example, V2A-Mapper proposes a mapping strategy that translates video features into audio CLAP embeddings, enabling AudioLDM to perform V2A synthesis. Similarly, FoleyCrafter integrates dedicated adapters to incorporate visual cues into the conditioning process of T2A models.

The second category includes methods such as TiVA (Wang et al., 2024b), ReWaS (Jeong et al., 2024), SyncFusion (Comunità et al., 2024), and SonicVisionLM (Xie et al., 2024), which incorporate explicitly designed temporal features to enhance synchronization between video and audio. TiVA employs downsampled Mel spectrograms as auxiliary representations that carry temporal structure, and utilizes a transformer-based predictor to estimate these features for guiding V2A generation. SyncFusion and SonicVisionLM leverage onset positions and audio timestamps, respectively, as temporal control signals during synthesis. ReWaS introduces energy as a continuous temporal representation, providing a more fine-grained condition along the time axis to better regulate V2A output. At the sampling stage, CFG is popularly employed in these methods. In our paper, we explore a novel sampling algorithm to increase generation quality in a training-free and computationally lightweight manner, which is orthogonal to previous innovations.

## H.3 GUIDANCE METHODS

**CFG.** In previous diffusion-based T2A generation (Liu et al., 2023; 2024a; Huang et al., 2023b;a; Evans et al., 2024; 2025; Lee et al., 2025) and V2A generation works (Sheffer & Adi, 2023; Luo et al., 2023; Mei et al., 2024; Xu et al., 2024; Wang et al., 2024a; Su et al., 2024; Liu et al., 2024b) , CFG (Ho & Salimans, 2022) is commonly adopted to improve the audio generation quality at the inference stage. To achieve optimal results, its guidance scale is investigated in different methods (Hung et al., 2024). However, as demonstrated in recent work (Lee et al., 2025), CFG sacrifices the diversity of generation results and may suffer from suboptimal synthesis quality. Previous theoretical analysis (Chidambaram et al., 2024) demonstrates that for any non-zero level of score estimation error, a large CFG strength causes the sampler to diverge from the data distribution's support, formally explaining the empirical

phenomenon of distortion at high guidance scales. Our analysis reveals that when CFG uses a bad unconditional model, it inherently introduces a score correction term. This may explain the empirical finding that a small unconditional model can effectively guide a large conditional model (Karras et al., 2024b). However, in standard CFG, this beneficial correction term is entangled with the conditional alignment term, making it difficult to find a guidance strength that simultaneously ensures outlier removal and quality enhancement.

**Guidance with weak models.** Recently, AG (Karras et al., 2024a) proposes a method to guide a diffusion model with the bad version of itself, demonstrating stronger synthesis quality than CFG in image domain. Several works (Phunyaphibarn et al., 2025; Jeon, 2025; Hyung et al., 2025) has extended this idea to the scenarios of fine-tuning diffusion models to a specific task (Phunyaphibarn et al., 2025), motion synthesis (Jeon, 2025), and video generation (Hyung et al., 2025). Other works adopt similar ideas to construct a weak model. For instance, SAG (Hong et al., 2023) applies Gaussian blurring to the model input, SG (Li et al., 2025) alters the denoising timestep to obtain an output with higher noise levels, while ICG (Sadat et al., 2024) randomly samples a condition to replace the null embedding in CFG. Among these guidance methods, AG is the most representative. It provides a theoretical justification for its efficacy, positing that it reduces the score estimation error induced by the "mass-covering" or "mean-seeking" behavior of the score matching training objective. However, the advantages of AG have not been observed for audio generation. As recently mentioned in ETTA (Lee et al., 2025), AG is sensitive to the choice of the bad model, prohibiting their application on audio synthesis. In this work, we explore the advantages of AG for audio generation, and propose a novel sampling algorithm, MoG, yielding cumulative advantages of AG and CFG to outperform either of them on audio synthesis.

## I   MORE GENERATED SAMPLES

We shown more text-to-audio and video-to-audio generation results in Figure 8 and 9, respectively. As shown in Figure 8, HG and PG consistently achieve more accurate harmonic modeling and superior temporal alignment compared to CFG in the first example. Their outputs exhibit well-defined harmonic stacks and consistent overtone structures, even in complex or polyphonic cases in the third example. Moreover, HG and PG maintain precise timing across events, effectively capturing the onset and duration of audio elements in the second example. In contrast, CFG often fails to organize harmonics coherently and produces temporally smeared results. These comparisons clearly illustrate the advantage of our methods in reinforcing both spectral clarity and temporal fidelity. In Figure 9, HG generates significantly clearer high-frequency content and overall higher-quality audio compared to CFG in the first example. The resulting audio exhibits more natural brilliance and detail in the upper frequency range, enhancing perceptual realism. For the second example, HG demonstrates superior temporal alignment, accurately synchronizing audio events with visual cues, while CFG shows noticeable temporal drift and inconsistent timing. These comparisons further highlight the strengths of our method in improving both spectral resolution and temporal coherence in V2A generation. For more generated samples, please refer to our demo page: `audiomog.github.io`.

## J   BROADER SOCIETAL IMPACT

Generative audio modeling presents substantial potential for misuse, which could result in harmful societal consequences. Principal concerns involve the dissemination of disinformation and the reinforcement of stereotypes and existing biases. Although our improvements enhance the realism and quality of generated samples, thereby potentially making misuse more convincing, they do not introduce any new capabilities or applications beyond those that already exist.

## K   LICENSES

- EDM2 models (Karras et al., 2024b;a): Creative Commons BY-NC-SA 4.0 license
- Stable Audio Tools (Evans et al., 2024): MIT license
- AudioLDM-Eval (Liu et al., 2023): MIT license
- CLIP (Radford et al., 2021): MIT license

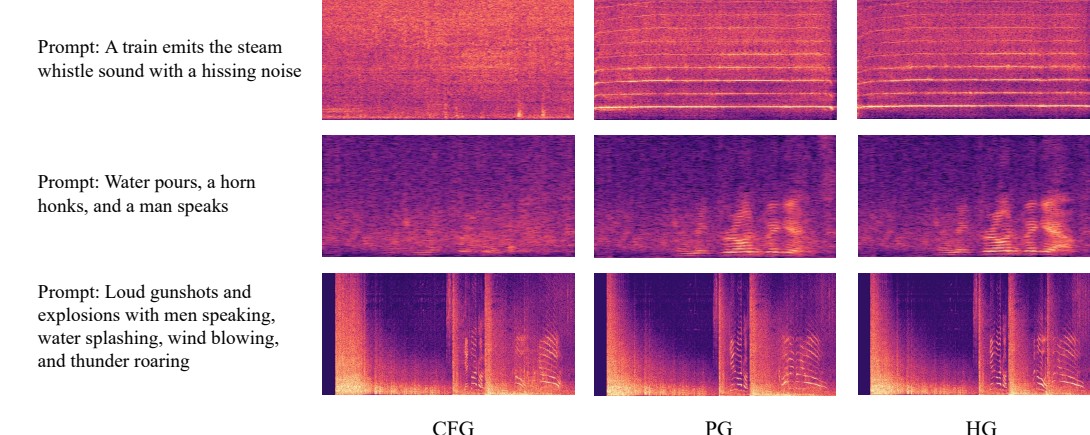

Prompt: A train emits the steam whistle sound with a hissing noise

Prompt: Water pours, a horn honks, and a man speaks

Prompt: Loud gunshots and explosions with men speaking, water splashing, wind blowing, and thunder roaring

CFG        PG        HG

Figure 8: **More T2A results comparing the spectrograms of the generated samples with different guidance strategies (CFG, PG, and HG) under various text prompts.** The third sample is shown with a different time interval than the one presented in the main paper, and they share the same text prompt.

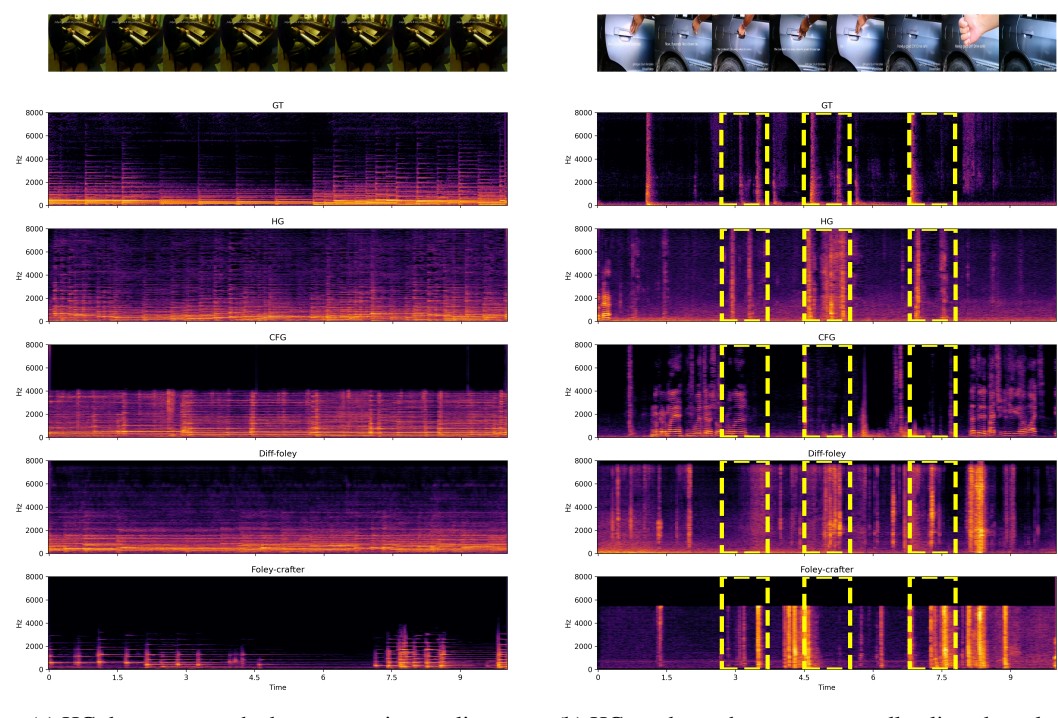

(a) HG demonstrates the best generation quality.      (b) HG produces the most temporally aligned results.

Figure 9: **More V2A comparing the spectrograms of the generated samples with different guidance strategies (CFG and HG) and baselines (Diff-foley and Foley-crafter).**

- Diff-foley models (Luo et al., 2023): Apache-2.0 license
- VTA-LDM models (Xu et al., 2024): Apache-2.0 license
- FoleyCrafter models (Zhang et al., 2024): Apache-2.0 license
- V2A-Mapper models (Wang et al., 2024a): Creative Commons BY-NC-ND 4.0 license

