# OpenReview forum: "AudioMoG: Guiding Audio Generation with Mixture-of-Guidance"
_ICLR.cc/2026/Conference — ICLR 2026 Conference Withdrawn Submission_

### Official Review · Reviewer_3hnZ · 2025-10-15

**Soundness:** 3
**Presentation:** 3
**Contribution:** 2
**Rating:** 4
**Confidence:** 3

**Summary:**

This work explores enriching the composition of the guidance method and presents a mixture-of-guidance framework, AudioMoG.

**Strengths:**

1. The mixture of guidance seems feasible and important.
2. The writing and visualization are good.
3. The results seem promising,

**Weaknesses:**

1. Evaluation: I’m not sure why there are so many “/” entries in Tables 1 and 2 for clearly open-source models (e.g., stable-audio-open). Reproducing the evaluation should be straightforward. It’s difficult to attribute any real performance gains relative to these baseline models.
2. Minor performance gains: For text-to-audio generation, the improvements appear quite small, typically around 0.01.
3. Motivation for mixture of guidance (MoG): MoG seems to be a general method applicable to diffusion-based generation across modalities. I don’t see what specifically motivates its application to audio versus images or other modalities. Does AudioMoG include any audio-specific adaptations?
4. Figure 2: It looks like AG performs as well as, or better than, HG. The statement “Hierarchical Guidance eliminates outliers and provides more controllable condition alignment” is unclear. Please clarify what “outliers” and “condition alignment” refer to and how they are measured.
5. Figure 3: The figure only shows curves for CFG and HG. Why not include AG and PG for a complete comparison?

**Questions:**

My concern is mainly about the comparison and the performance. And I will reconsider my score if they are well addressed.

---

### Official Review · Reviewer_MdQx · 2025-10-26

**Soundness:** 2
**Presentation:** 4
**Contribution:** 2
**Rating:** 2
**Confidence:** 5

**Summary:**

This work presents a mixture-of-guidance (MoG) framework that integrates classifier-free guidance (CFG) and auto guidance (AG) to improve text-to-audio, video-to-audio, text-to-music, and class conditional image generation performance. Through the experiments, the hierarchical guidance (MoG-HG) that the authors present and parallel guidance (MoG-PG) show better performance than CFG on objective metrics.

**Strengths:**

- Through extensive experiments, this work demonstrates that a linear combination of CFG and AG shows better performance compared to using only CFG or only AG.
- Furthermore, this improvements are confirmed across multiple tasks: text-to-audio, video-to-audio, text-to-music, and class conditional image generation.

**Weaknesses:**

In the proposed MoG, the hierarchical guidance (MoG-HG), which should be presented as the primary contribution, does not move beyond the framework of a linear combination of the scores used in CFG and AG. This limits it to being an incremental extension from exsisting framework. Furthermore, MoG-HG shows only a slight numerical improvement over MoG-PG.

Moreover, the MoG-PG appears to be merely a more detailed experimental study of tuning CFG and AG that reported in the appendix of ETTA [1]. The reviewer acknowledges that the comprehensive experiments validates that MoG-PG leads better results against CFG-only and AG-only baselines, but the contribution to the research field is limited.

Additionally, the paper does not fundamentally solve the non-trivial challenge of obtaining a degraded weak model for AG, a problem the authors point out in the Introduction and the "AG effects" paragraph of Section 3.1.

Finally, for MoG-HG, it would seem straightforward to apply it to open-sourced pretrained models, which the authors do in the image generation experiments by using EDM2. However, it is unclear why the authors chose to train a new model from scratch and validate the effectiveness of MoG-PG and MoG-HG only on that specific model on text-to-audio, video-to-audio, and text-to-music generation experiments.

[1] Lee, Sang-gil, et al. "ETTA: Elucidating the Design Space of Text-to-Audio Models." ICML 2025.

**Questions:**

- Can MoG-HG demonstrate superior performance compared to its MoG-PG counterpart when applied to pretrained models like AudioLDM2 (or AudioLDM), Stable Audio Open, ETTA, and MMAudio [2]? Corresponding potential weak degraded models exist for the AudioLDM series, Stable Audio Open, ETTA, and MMAudio. If the authors can show that MoG-HG outperforms MoG-PG using these respective weak models, it would substantially strengthen the experimental validity of the proposed method.

[2] Cheng, Ho Kei, et al. "MMAudio: Taming Multimodal Joint Training for High-Quality Video-to-Audio Synthesis." CVPR 2025

---

### Official Review · Reviewer_RNen · 2025-10-31

**Soundness:** 3
**Presentation:** 1
**Contribution:** 2
**Rating:** 2
**Confidence:** 4

**Summary:**

This paper presents comprehensive analysis on the impact of the inference time guidance for diffusion-based audio generative models. More specifically,
- Task: Text2audio (TTA) and Video2audio (Foley). NOTE: Text2music and image generation results are only presented in Appendix.
- Audio generative backbone: a newly trained model based on the Stable-audio-open architecture
- type of guidances: Classifier-free (CFG) and Autoguidance (AG)
- Contribution: Verifying that
    1. The combination of CFG and AG improves audio generation performance
    2. (Not clear from the paper, just my guess) A checkpoint saved at earlier training steps (e.g., a checkpoint saved at 100k steps when the full training is 1M steps) can serve as the bad model

I believe the work presents findings that are beneficial to the audio generation community. However, the contributions and findings are within a very limited scope and seem to be incremental. For example, in StemGen https://arxiv.org/abs/2312.08723, authors present a new method to achieve stem-wise music generation, and to improve the performance, they proposed to combine Multiple CFG. In audio domain, the combination of multiple guidance is usually **a part of a new method**, but NOT a new method itself.

The authors seem to have chosen out-of-date methods to compare with, such that this paper cannot present the frontier in audio generation community.

As a reviewer, I tend to reject this paper.

**Strengths:**

- Showed that the combination of CFG and AG at inference time enhances audio generation performance

**Weaknesses:**

## Limited Scope
- The contributions and findings are within a very limited scope and seem to be incremental. In audio domain, the combination of multiple guidance is usually **a part of a new method**, but NOT a new method itself. For example,
- in StemGen https://arxiv.org/abs/2312.08723, authors present a new method to achieve stem-wise music generation, and to improve the performance, they proposed to combine Multiple CFG.
- Similarly, see that multiple guidance is a part of JASCO https://arxiv.org/abs/2406.10970
## Potentially inaccurate theoretical description
- According to Appendix F.3 and F.4, the audio generation backbone seems to be very similar to Stable-Audio-Open
    - Stable-Audio-Open uses v-diffusion, where the training target is "velocity"
    - In Section 2 formula (2), I don't see "velocity".
    - Please clarify which training loss is used
- In formula (8), the summation of weights should be 1.
    - In the remaining part of this paper, obvious this constraint is not used. Please clarify if this constraint is actually used or not.
## Out-of-date models in Table1 text2audio
- It is strange to exclude Make-an-audio-2 from the table, when Make-an-audio-1 is included.
    - The FAD of Make-an-audio-1 seems to be wrong. In Make-an-audio-2 paper, the FAD for ver.1 is 2.66 not 1.61.
- Methods published in 2025, such as SoundCTM https://arxiv.org/abs/2405.18503, AudioTurbo https://arxiv.org/abs/2505.22106, are ingored.
- Stable-Audio-Open is open-source, but the authors leave many metrics blank. These metrics can be measured by inferrring with the open model weight.
## Out-of-date models, less proper metrics, and less proper model design in Table3 video2audio foley task
- MMAudio is ignore, which is hard to understand, if we consider the impact of this model.
- Since the foley function of AudioMoG is achieved by finetuning the TTA version, it would be good to mention other methods that are also built upon a pretrained TTA model, such as CAFA https://arxiv.org/abs/2504.06778 or SpecMaskFoley https://arxiv.org/abs/2505.16195
- The evaluation in table3 is based on DiffFoley, an out-of-date model.
    - The DeSync metric used in MMAudio can be a better candidate. For example, Align acc. says DiffFoley and FoleyCrafter is BETTER-THAN-GT, which does match their actual performance, see MMAudio demo page https://hkchengrex.com/MMAudio/video_main.html#
    -   Meanwhile, DeSync metrics used in MMAudio and SpecMaskFoley seems to align better with human perception
- While AudioMoG uses CLIP for video synchronization, it has already been shown ineffective in prior works. V-AURA, Multi-Foley, MMAudio, and SpecMaskFoley all showed that other video features can be a better alternative

**Questions:**

## Potentially inaccurate theoretical description
- According to Appendix F.3 and F.4, the audio generation backbone seems to be very similar to Stable-Audio-Open
    - Stable-Audio-Open uses v-diffusion, where the training target is "velocity"
    - In Section 2 formula (2), I don't see "velocity".
    - Please clarify which training loss is used
- In formula (8), the summation of weights should be 1.
    - In the remaining part of this paper, obvious this constraint is not used. Please clarify if this constraint is actually used or not.

---

### Official Review · Reviewer_1Maf · 2025-11-01

**Soundness:** 3
**Presentation:** 3
**Contribution:** 1
**Rating:** 4
**Confidence:** 4

**Summary:**

This paper proposes AudioMoG, a mixture-of-guidance (MoG) sampling framework for cross-modal audio generation (T2A, V2A), with additional results on text-to-music and conditional image generation. The central idea is to combine two established guidance families—classifier-free guidance (CFG) and autoguidance (AG)—either hierarchically (HG) or in parallel (PG). The authors argue that CFG primarily boosts condition alignment but can reduce diversity, while AG improves score estimation via a weak/“bad” model. They show (i) a simple linear-combination view that subsumes CFG/AG and a hierarchical combination that yields “cumulative benefits,” (ii) a small theoretical equivalence (embedding AG into CFG vs. CFG into AG), and (iii) empirical gains under matched inference cost (NFE) on AudioCaps and VGGSound, plus T2M and ImageNet-512. The motivation aligns with recent analyses of guidance trade-offs and the AG line of work.

**Strengths:**

1. Training-free and practical: Improves quality at sampling time without retraining.

2. Consistent empirical gains under matched NFE across T2A and V2A, with additional evidence on T2M

**Weaknesses:**

Marginal conceptual novelty. The core mechanism is essentially weighted combinations of existing guidance signals (CFG terms and weak/strong contrasts) with a hierarchical schedule. This overlaps with prior AG and CFG reinterpretations that already discuss predictor/corrector views and weak-model contrasts; AudioMoG’s novelty is mostly in the composition policy and empirical validation on audio.

**Questions:**

How does a single set of (w_1, w_2, w_3) transfer across datasets without re-tuning?

---

### Note · Authors · 2025-12-08

I have read and agree with the venue's withdrawal policy on behalf of myself and my co-authors.